# Electrochemically active bacteria sense electrode potentials for regulating catabolic pathways

Atsumi Hirose[1], Takuya Kasai[1], Motohide Aoki[1], Tomonari Umemura[1], Kazuya Watanabe[1] &
Atsushi Kouzuma [1]

Electrochemically active bacteria (EAB) receive considerable attention for their utility in bioelectrochemical processes. Although electrode potentials are known to affect the metabolic activity of EAB, it is unclear whether EAB are able to sense and respond to electrode potentials. Here, we show that, in the presence of a high-potential electrode, a model EAB *Shewanella oneidensis* MR-1 can utilize NADH-dependent catabolic pathways and a background formate-dependent pathway to achieve high growth yield. We also show that an Arc regulatory system is involved in sensing electrode potentials and regulating the expression of catabolic genes, including those for NADH dehydrogenase. We suggest that these findings may facilitate the use of EAB in biotechnological processes and offer the molecular bases for their ecological strategies in natural habitats.

---

[1] School of Life Sciences, Tokyo University of Pharmacy and Life Sciences, 1432-1 Horinouchi, Hachioji, Tokyo 192-0392, Japan. Correspondence and requests for materials should be addressed to A.K. (email: akouzuma@toyaku.ac.jp)

Bioelectrochemical systems (BES) are engineered systems in which electrochemically active bacteria (EAB) are grown under electrochemical interactions with electrodes[1]. These systems have attracted wide attention owing to their utility in various biotechnological processes, including the generation of electric power from organic waste (e.g., microbial fuel cells)[2] and the production of commodities and fine chemicals using electricity as the sole energy source (e.g., microbial electrosynthesis)[3]. In BES, electrodes serve as electron acceptors or donors for EAB[1]. A number of studies have shown that electrode potentials primarily determine the direction and rate of electron flow between electrodes and EAB, thereby influencing the metabolic activity of EAB[4–8]. Moreover, the electrode potential thermodynamically limits the maximum amount of energy that EAB can conserve in BES[9]. However, it is unclear whether or not EAB can actively respond to electrode potentials and regulate catabolic pathways for energy conservation.

Recent studies have suggested that catabolic and respiratory pathways in EAB are changed in response to shifts in electrode potentials. Ishii et al. have reported that a high-potential electrode stimulates the expression of respiratory genes, including those for outer membrane (OM) c-type cytochromes, in electrode-associated microbial communities[10]. Transcriptome and proteome analyses have shown that *Shewanella oneidensis* MR-1 and *Shewanella loihica* PV-4 exhibit altered expression of genes for the tricarboxylic acid (TCA) cycle and lactate metabolism when they are grown at different electrode potentials[11–13]. Furthermore, *Geobacter sulfurreducens* expresses different inner membrane (IM)-localized respiratory cytochromes depending on the electrode potential[14,15]. These findings suggest that EAB may be able to link intracellular catabolic and respiratory pathways to electrode potentials.

*S. oneidensis* MR-1 is the representative strain of the genus *Shewanella* and is one of the most extensively studied EAB due to its annotated genome sequence, ease of cultivation and genetic manipulation, and ability to transfer electrons to extracellular electrodes without the need for an exogenous mediator[16,17]. Genetic and biochemical studies have revealed that the extracellular electron transfer (EET) pathway in MR-1 consists of an IM-anchored cytochrome (CymA), soluble periplasmic cytochromes (STC and FccA) and an OM cytochrome complex (comprised of MtrA, MtrB, OmcA and MtrC)[16–18]. Studies have also demonstrated that MR-1 has a well-developed respiratory network consisting of periplasmic and membrane-bound electron-transfer proteins for efficiently discharging electrons to various organic and inorganic electron acceptors (e.g., oxygen, fumarate, nitrate, thiosulfate, trimethylamine *N*-oxide, dimethylsulfoxide, and solid-phase electron acceptors, such as electrodes and metal oxides)[16–20]. This respiratory versatility is thought to facilitate the survival of *Shewanella* in redox-stratified natural habitats, where available electron acceptors are limited and/or frequently changed (e.g., oxic/anoxic interfaces in sediments)[16]. Since these electron acceptors have their own redox potentials, it is likely that they have evolved the ability to efficiently conserve energy according to the redox potential of electron acceptors. Despite extensive characterization of electron-transfer pathways, however, it remains unclear how redox potentials of electron acceptors affect the growth of MR-1. We consider that BES are useful for examining this question, because these systems allow for the fine control of redox potentials of electron acceptors (i.e., electrodes).

Based on current knowledge of EAB, we hypothesized that EAB harbor molecular mechanisms for sensing electrode potentials and regulating catabolic pathways. To address this hypothesis, we examined *S. oneidensis* MR-1 grown in BES under different electrode potentials and demonstrate that MR-1 has

molecular mechanisms for electrode potential-dependent catabolic regulation. The findings presented here not only provide the molecular basis for the biotechnological application of EAB in BES but also offer molecular insights into the ecological strategies of EAB in their natural habitats.

## Results

**Responses of MR-1 to different electrode potentials.** *S. oneidensis* MR-1 was grown in an electrochemical cell (EC) equipped with a working electrode poised at +0.5 V (high potential; HP), +0.2 V (middle potential; MP), or 0 V (low potential; LP) (vs. the standard hydrogen electrode; SHE) with lactate as the electron donor. Potential values are reported in reference to SHE. Current densities, metabolite concentrations, and protein yields were compared among HP, MP, and LP conditions (Fig. 1). Higher electric currents were observed under higher potential conditions (Fig. 1a), indicating that high electrode potentials facilitate current generation. Once the electric currents had dropped, culture supernatants were collected for metabolite analyses. While lactate was not detected in these EC samples, acetate was detected as the major metabolite (Fig. 1b), indicating that the partial oxidation of lactate to acetate was the major electron source for EET, as reported elsewhere[13,21]. We also found that the molar yield of acetate per lactate was significantly lower under the HP condition than the LP condition (Fig. 1b). In addition, protein analyses revealed that growth under the HP condition resulted in a higher protein yield (per lactate consumed) than that under the LP condition (Fig. 1c). The acetate and protein yields under the MP condition were moderate and between those for the HP and LP conditions. These results suggest that MR-1 can conserve more

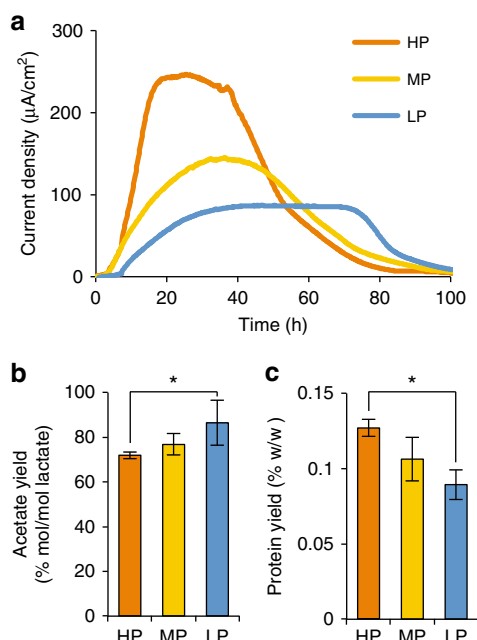

**Fig. 1** Response of *S. oneidensis* MR-1 to different electrode potentials. **a** Current generation from lactate in electrochemical cells (EC) operated with high potential (HP; +0.5 V), middle potential (MP; +0.2 V), or low potential (LP; 0 V) electrodes. Results represent the mean of three independent experiments (*n* = 3 biological replicates). **b**, **c** Acetate (**b**) and protein (**c**) yields from lactate under HP and LP conditions. Bars and error bars represent means and standard deviations (s.d.), respectively (*n* = 3 biological replicates). Asterisks indicate a statistically significant difference (*P* < 0.05; one-way ANOVA followed by LSD test)

energy using the HP electrode than the LP electrode. It is also suggested that catabolic pathways in MR-1 can be changed in response to electrode potentials.

**Transcriptomic responses to different electrode potentials**. To explore how *S. oneidensis* MR-1 responds to different electrode potentials at the molecular level, the bacterium was exposed to two different electrode potentials (+0.5 V (HP) and −0.1 V (LP); current vs. time curves are shown in Supplementary Figure 1), and their transcriptomic profiles at these potentials were analyzed using DNA microarrays. We choose these two potential conditions because comparison of the microarray data obtained under the HP and MP (+0.2 V) conditions or the MP and LP conditions did not identify any differentially expressed genes with statistical significance (fold change ≥2.0 or ≤0.5, *P* value <0.05; paired Student's *t* test followed by Benjamini−Hochberg false discovery rate correction). However, the microarray data obtained under the HP and LP conditions showed that the expression of 322 genes was significantly altered by the potential shift. Among these electrode potential-responsive (EPR) genes, 274 were upregulated, while 48 were downregulated in the presence of the HP electrode (Supplementary Data 1). The reliability of the microarray analyses was validated by quantitative reverse transcription PCR (qRT-PCR; Supplementary Figure 2). Many of the downregulated genes encode proteins with unknown functions, whereas a substantial number of the upregulated genes have been assigned to categories of the Clusters of Orthologous Groups of proteins[22], in particular the "Energy production and conservation [C]" category

(Supplementary Figure 3). Among these upregulated genes were those for D-lactate dehydrogenase (*dld*), NADH:ubiquinone oxidoreductase (*nuo*), and $F_0F_1$ ATP synthase (*atp*) (Table 1), suggesting that HP electrodes activate important catabolic and energy-conserving pathways in MR-1.

Upregulation of the *nuo* and *atp* genes in the presence of the HP electrode is intriguing, because a previous study reported that, under anaerobic conditions (e.g., fumarate-respiring conditions), MR-1 generates ATP by substrate-level phosphorylation at the acetate-synthesis step (Fig. 2), and oxidative phosphorylation using respiratory chains and $F_0F_1$ ATP synthase does not significantly contribute to energy conservation[23]. Other studies have also shown that anaerobic lactate catabolism in MR-1 does not require NADH-dependent enzymes, such as pyruvate dehydrogenase (PDH), NADH dehydrogenase (NDH) and other NAD-dependent dehydrogenases in the TCA cycle[23–29]. Our results, however, indicate that NADH oxidation and ATP generation are upregulated during respiration using the HP electrode. Together with our findings that genes encoding enzymes involved in NADH regeneration, such as PDH (*aceF*) and 2-oxoglutarate dehydrogenase (*sucAB*), were upregulated under the HP condition (Table 1), our analyses suggest that NADH-dependent catabolic pathways are activated in the presence of an HP electrode.

We also found that, among the three gene clusters encoding formate dehydrogenases (FDH), SO_101–SO_103 and SO_4513–SO_4515 were upregulated while SO_4509–SO_4511 was downregulated under the HP condition. SO_101–SO_103

**Table 1 List of selected electrode potential-responsive genes discussed in this study**

| Process | Locus tag | Gene | Annotation | Log$_2$ FC[a] |
|---|---|---|---|---|
| Lactate and pyruvate oxidation | SO_1521 | *dld* | Respiratory FAD-dependent D-lactate dehydrogenase | 2.32 |
| | SO_0425 | *aceF* | Dihydrolipoamide acetyltransferase | 1.52 |
| Formate oxidation | SO_0101 | *fdnG* | Nitrate-inducible formate dehydrogenase molybdopterin-binding subunit | 2.97 |
| | SO_0102 | *fdnH* | Nitrate-inducible formate dehydrogenase iron-sulfur subunit | 3.33 |
| | SO_0103 | *fdnI* | Nitrate-inducible formate dehydrogenase cytochrome b subunit | 2.89 |
| | SO_4509 | *fdhA* | Formate dehydrogenase molybdopterin-binding subunit | −1.43 |
| | SO_4510 | *fdhB* | Formate dehydrogenase fes subunit | −1.08 |
| | SO_4511 | *fdhC* | Formate dehydrogenase cytochrome b subunit | −1.00 |
| | SO_4513 | *fdhA* | Fnr-inducible formate dehydrogenase molybdopterin-binding subunit | 1.96 |
| | SO_4515 | *fdhC* | Fnr-inducible formate dehydrogenase cytochrome b subunit | 1.94 |
| TCA cycle | SO_1930 | *sucA* | 2-Oxoglutarate dehydrogenase complex dehydrogenase E1 component | 1.75 |
| | SO_1931 | *sucB* | 2-Oxoglutarate dehydrogenase complex succinyl-CoA:dihydrolipoate S-succinyltransferase E2 component | 1.77 |
| | SO_1933 | *sucD* | Succinyl-CoA synthase alpha subunit | 1.66 |
| NADH oxidation | SO_1010 | *nuoM* | NADH-ubiquinone oxidoreductase subunit M | 2.85 |
| | SO_1012 | *nuoK* | NADH-ubiquinone oxidoreductase subunit K | 2.68 |
| | SO_1013 | *nuoJ* | NADH-ubiquinone oxidoreductase subunit J | 2.58 |
| | SO_1014 | *nuoI* | NADH-ubiquinone oxidoreductase subunit I | 2.46 |
| | SO_1015 | *nuoH* | NADH-ubiquinone oxidoreductase subunit H | 2.89 |
| | SO_1016 | *nuoG* | NADH-ubiquinone oxidoreductase subunit G | 2.61 |
| | SO_1017 | *nuoF* | NADH-ubiquinone oxidoreductase subunit F | 2.46 |
| | SO_1018 | *nuoE* | NADH-ubiquinone oxidoreductase subunit E | 1.78 |
| | SO_1019 | *nuoCD* | NADH-ubiquinone oxidoreductase subunit CD | 1.90 |
| ATP synthesis | SO_4746 | *atpC* | ATP synthase F1 epsilon subunit | 2.19 |
| | SO_4747 | *atpD* | ATP synthase F1 beta subunit | 2.22 |
| | SO_4748 | *atpG* | ATP synthase F1 gamma subunit | 2.50 |
| | SO_4749 | *atpA* | ATP synthase F1 alpha subunit | 2.43 |
| | SO_4750 | *atpH* | ATP synthase F1 delta subunit | 1.94 |
| | SO_4751 | *atpF* | ATP synthase F0 B subunit | 1.85 |
| | SO_4752 | *atpE* | ATP synthase F0 C subunit | 1.71 |
| | SO_4753 | *atpB* | ATP synthase F0 A subunit | 1.37 |

[a] Log$_2$-transformed fold change (+0.5 V/−0.1 V)

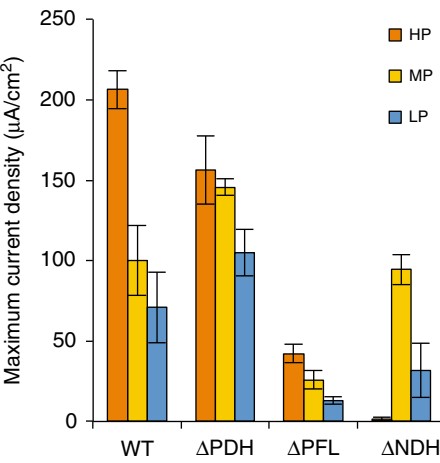

**Fig. 2** Catabolic pathways of lactate to acetate in *S. oneidensis* MR-1. LDH lactate dehydrogenase, Q oxidized form of quinone, QH2 reduced form of quinone, NDH NADH dehydrogenase, PDH pyruvate dehydrogenase, PFL pyruvate formate-lyase, PTA phosphotransacetylase, AK acetate kinase

**Fig. 3** Current generation from pyruvate by WT and mutant *S. oneidensis* strains. The maximum current densities at +0.5 V (HP), +0.2 V (MP), or 0 V (LP) are shown. Bars and error bars represent means and s.d., respectively ($n = 3$ biological replicates)

(*fdnGHI*) is predicted to encode a selenocysteine-containing FDH similar to the nitrate-inducible Fdh-N in *Escherichia coli*[30,31]. SO_4509–SO_4511 and SO_4513–SO_4515 are both annotated as *fdhABC* and are homologous to the genes encoding Fdh-O in *E. coli*[31]. Although the results demonstrate that MR-1 differentially regulates these FDHs according to the electrode potential, triggering factors and molecular mechanisms for the transcriptional regulation are currently unknown.

**Electrode potential-dependent shift of catabolic pathways**. From our transcriptomics data, we hypothesized that for pyruvate oxidation MR-1 specifically uses an NADH-dependent catabolic pathway comprised of PDH and NDH under HP conditions, despite that it also has a formate-dependent catabolic pathway comprised of pyruvate formate-lyase (PFL) (Fig. 2). To examine this hypothesis, we generated deletion mutants of PDH (ΔPDH) and PFL (ΔPFL) and investigated the contribution of these enzymes to current generation under HP (+0.5 V), MP (+0.2 V), and LP (0 V) conditions. To directly observe the contribution of the two splitting pathways to pyruvate oxidation under the three electrode-potential conditions, we used pyruvate as the electron donor for current generation. Comparison of the maximum current densities generated by these mutants (Fig. 3) revealed that, under the HP condition, ΔPDH and ΔPFL generated approximately 20 and 80% less current, respectively, than wild-type MR-1 (WT). This result demonstrates that PDH contributes to pyruvate catabolism under HP conditions, albeit to a lesser extent than PFL. In contrast, under the MP and LP conditions, ΔPDH generated more current while ΔPFL generated markedly less current than WT, demonstrating that pyruvate catabolism under these relatively low potential conditions mostly depends on PFL but not PDH. These results also suggest that the presence of PDH negatively affects catabolism under these potential conditions where the formate-dependent catabolic pathway is predominant. For comparison, we also examined current generation by a *cymA*-knockout mutant (Δ*cymA*)[32]. The Δ*cymA* strain generated only negligible amounts of current under the HP and LP conditions (Supplementary Figure 4), confirming that MR-1 utilizes this cytochrome for current generation regardless of the electrode potential.

We also investigated the involvement of NDH in current generation from pyruvate. Since the MR-1 genome encodes four NDH complexes, Nuo, Nqr1, Nqr2, and Ndh[33,34], we generated a deletion mutant of the four NDH complexes (ΔNDH) and

measured the current generated by this mutant under HP, MP, and LP conditions (Fig. 3). While ΔNDH almost completely lost the ability to generate current under the HP condition, it was able to generate current under the MP and LP conditions. We hypothesize that there may be excessive accumulation of NADH in MR-1 cells under the HP condition, resulting in the inhibition of some oxidative metabolic reactions and cell growth[35]. To investigate this possibility, we calculated the intracellular NADH/NAD+ ratio in MR-1 cells, and confirmed that the deletion of NDHs resulted in a higher NADH/NAD+ ratio compared to WT under the HP condition (Supplementary Figure 5). Although NADH was also accumulated in ΔNDH under the MP and LP condition, the NADH/NAD+ ratios were decreased as the electrode potential was lowered (Supplementary Figure 5). Since ΔNDH retained the ability to generate current under the MP condition (Fig. 3), it is suggested that there is a threshold NADH/NAD+ ratio that inhibits essential cellular processes for current generation. Taken together, these results support the idea that the NADH-dependent pathway is involved in pyruvate catabolism under HP conditions. However, given that the contribution of PDH was relatively small under the HP condition (approximately 20%; Fig. 3), we conclude that MR-1 primarily uses the formate-dependent pathway for current generation and activates the NADH-dependent pathway under HP conditions.

**Involvement of NDH in manganese-oxide reduction**. To investigate whether or not MR-1 alters pyruvate-catabolic pathways according to available electron-acceptor compounds, WT and ΔNDH strains were cultivated using pyruvate as the electron donor in the presence of one of the following electron acceptors with different standard redox potentials ($E_0$ at pH 7.0): oxygen (+0.82 V), $MnO_2$ (+0.53 V), nitrate (+0.43 V), and fumarate (−0.03 V)[36,37]. Under the fumarate-reducing condition, ΔNDH showed a similar growth rate to WT (Fig. 4a), suggesting that MR-1 does not utilize the NADH-dependent catabolic pathway for reducing electron acceptors with redox potentials around 0 V. However, ΔNDH showed a decreased ability to use nitrate (Fig. 4b) and $MnO_2$ (Fig. 4c), and did not grow under aerobic conditions (Fig. 4a). These results are in agreement with our observation that ΔNDH generated very little current under the HP condition, and supports the idea that MR-1 uses electron acceptors with relatively high redox potentials by activating the NADH-dependent catabolic pathway.

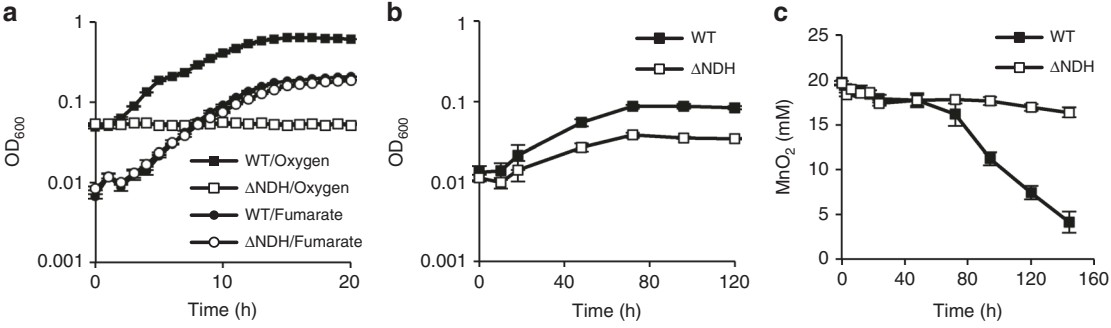

**Fig. 4** Growth characteristics of ΔNDH on different electron acceptors. **a**, **b** Growth curves of WT and ΔNDH cells grown on pyruvate under aerobic, fumarate-reducing (**a**), and nitrate-reducing (**b**) conditions. **c** Reduction of $MnO_2$ by WT and ΔNDH cells. Data points and error bars represent means and s.d., respectively ($n = 3$ biological replicates)

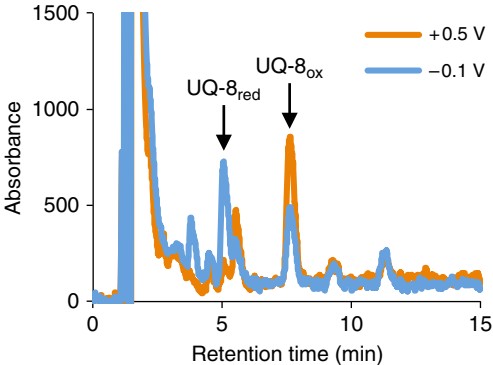

**Fig. 5** Detection of UQ-8 in MR-1 cells exposed to different electrode potentials. The retention times of oxidized and reduced ubiquinones (UQ-$8_{ox}$ and UQ-$8_{red}$, respectively) are indicated by arrows

**Electrode-potential sensing by the Arc regulatory system.** Our results so far reveal that MR-1 regulates catabolic pathways in response to changes in the redox potential of extracellular insoluble electron acceptors (e.g., electrodes) as well as soluble electron acceptors. We were therefore interested in investigating how MR-1 perceives the potential of an electrode that is located outside of cells. Bacteria are known to regulate the expression of cellular processes by sensing redox states of intracellular electron mediators, such as IM quinones[38]. As IM quinones are connected to electrodes via the EET pathway[18,39], we postulated that an electrode potential affects the redox state of IM quinones, and that MR-1 senses changes in the redox state using an intracellular redox sensor, such as the Arc system[40,41]. To investigate this hypothesis, we first examined whether the redox balance of IM quinones is affected by electrode potentials, that is, by HP and LP. As our transcriptome analysis showed that the expression of *nuo* genes encoding NADH:ubiquinone oxidoreductase was upregulated under the HP condition (Table 1), we focused on detecting the potential-dependent redox shift of ubiquinone, even though ubiquinone with its relatively high redox potential (+0.11 V) is generally used as an IM electron carrier for aerobic respiration[37]. A previous study reported that the major IM quinone in MR-1 cells is ubiquinone-8 (UQ-8)[20]; therefore, we examined the oxidized and reduced forms of UQ-8 in cells exposed to different electrode potentials (+0.5 V and −0.1 V) (Fig. 5). Our analysis revealed that the presence of the HP electrode increased the oxidized form of UQ-8 (Fig. 5), demonstrating that the redox balance of the IM quinone pool is affected by electrode potential. This result also indicates that MR-1 utilizes UQ-8 as an IM electron carrier for current generation.

We next examined the involvement of the Arc regulatory system, which is known to serve as a sensor for the redox state of IM quinones in *E. coli*[40,41], in electrode-potential sensing by MR-1. The Arc system of MR-1 consists of three components, the sensor histidine kinase (ArcS), the phosphotransfer protein (HptA), and the response regulator (ArcA)[42]. We generated a deletion mutant of the sensor kinase gene (ΔarcS) and examined current generation and the transcriptome profiles of this mutant under potential-controlled conditions (+0.5 V and −0.1 V). We found that ΔarcS generated markedly lower current under these potential conditions (Supplementary Figure 1) compared to WT, demonstrating that the Arc system is involved in current generation in MR-1. Transcriptome analysis of ΔarcS cells grown under these conditions revealed that, among the 322 EPR genes found in WT, 305 genes, including the *nuo* and *atp* genes, did not show significant transcriptional responses in ΔarcS (Fig. 6a). Further, 105 genes were responsive to the potential shift only in ΔarcS (Fig. 6a and Supplementary Data 2). However, mean average (MA) plots of the microarray data for WT and ΔarcS demonstrate overall decreases in the fold change in gene expression in ΔarcS compared to WT (Fig. 6b). These results demonstrate that the Arc system plays a central role in electrode potential-dependent transcriptional regulation in MR-1. qRT-PCR analysis showed that *nuoI* gene expression in WT was increased under the HP condition and decreased under the LP condition, and was decreased in ΔarcS regardless of the electrode potential (Supplementary Figure 6). The hierarchical clustering of expression patterns of the 322 EPR genes found in WT suggests that many of these genes exhibited similar expression patterns to the *nuoI* gene (Supplementary Figure 7), demonstrating that deletion of *arcS* results in decreased expression of many EPR genes. Microarray analysis also identified 17 genes that were responsive to the potential shift in both WT and ΔarcS (Fig. 6a and Supplementary Data 3). These ArcS-independent EPR genes included the gene encoding NADH-independent D-lactate dehydrogenase (*dld*). We previously reported that *dld* gene expression is increased under HP conditions[13] and positively regulated by CRP[43]; the present result indicates that the Arc system is not involved in the regulation of this gene.

We also performed electrophoretic mobility shift assays (EMSA) using purified ArcA protein (Supplementary Figure 8) to examine whether or not the Arc system directly regulates the transcription of *nuo* and other ArcS-dependent EPR genes. When phosphorylated ArcA (ArcA-P) was incubated with a DNA probe containing the upstream region of *nuoA*, shifted bands were observed in a protein concentration-dependent manner (Fig. 7). However, no shifted bands were detected when probes containing the upstream regions of *sucA* and *atpI* were used (Supplementary Figure 9). These results indicate that the Arc system directly

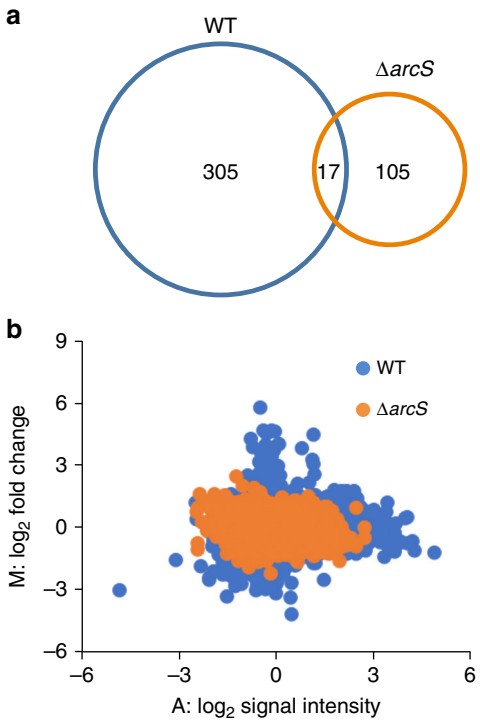

**Fig. 6** Electrode potential-dependent transcriptional changes in WT and $\Delta arcS$. **a** Venn diagram showing the number of electrode potential-responsive genes found in WT and $\Delta arcS$. **b** Mean average (MA) plots of microarray data for WT and $\Delta arcS$. M and A values indicate log$_2$-transformed fold changes and normalized signal intensities for each microarray probe, respectively

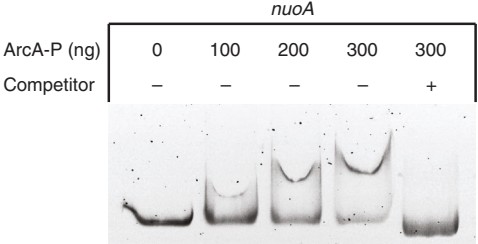

**Fig. 7** EMSA using ArcA protein and the upstream region of *nuoA*. The labeled probe containing the upstream region of *nuoA* was incubated with 0–300 ng phosphorylated ArcA (ArcA-P) in the presence (+) or absence (−) of a specific competitor (a 50-fold excess of unlabeled *nuoA* probe)

regulates the expression of the *nuo* genes, but indirectly affects the expression of the *suc* and *atp* genes.

## Discussion

We show that MR-1 perceives electrode potentials using the Arc system and regulates the expression of catabolic pathways (Fig. 8); in particular, it activates NADH-dependent pyruvate catabolism at high electrode potentials, resulting in a high growth yield (Fig. 1c). Under anaerobic conditions, including LP conditions, MR-1 catabolizes pyruvate using a formate-dependent pathway consisting of PFL and FDH (Fig. 8a), as observed for other anaerobic bacteria[24]. In these processes, the redox cycling of IM quinones catalyzed by FDH and CymA generates proton motive force (PMF)[26]. Under HP conditions, however, MR-1 activates the NADH-dependent pathway consisting of PDH and NDH for catabolizing a portion of pyruvate (Fig. 8b). This catabolic pathway is similar to that used by MR-1 for aerobic respiration[27]. Activation of Nuo NDH increases the amount of PMF generated per electron (H$^+$/e$^-$ ratio), since this enzyme also functions as an efficient proton pump[44]. The H$^+$/e$^-$ ratios of NADH oxidation by Nuo and formate oxidation by FDH are two and one, respectively[30,44,45]. Given that the redox cycling of ubiquinone between Nuo and CymA also contributes to PMF generation, MR-1 can generate an estimated three times more PMF per electron when it catabolizes pyruvate to acetyl-CoA and CO$_2$ using the NADH-dependent pathway compared to that using the formate-dependent pathway (Fig. 8). In addition, our metabolite and transcriptome analyses (Fig. 1b and Table 1) suggest that the TCA cycle is partially activated under HP conditions, although it has a smaller contribution to acetyl-CoA catabolism than the acetate-synthesis pathway. These findings indicate that MR-1 can efficiently generate PMF and conserve energy by regulating intracellular catabolic pathways according to the redox potential of electron acceptors (e.g., electrodes). Since the electromotive force (EMF) of biological energy conservation is mainly determined by redox potentials of electron acceptors, the Arc-regulated catabolic system may constitute a mechanism that links EMF to PMF.

We showed that MR-1 uses the NADH-dependent catabolic pathway for respiration using MnO$_2$ (Fig. 4c). This finding suggests that MR-1 can conserve more energy by respiration using MnO$_2$ than using electron acceptors with low redox potentials. Considering that MR-1 was isolated from the sediments of a shallow eutrophic lake containing iron/manganese-oxide concretion at the bottom[19], it is likely that this strain has adapted to anaerobic organics-rich environments with a variety of electron acceptors, including metal oxides[36]. In addition, exterior redox states may frequently change due to geochemical and biological events, including gravitational metal-oxide settlings and algal blooms[36]. We therefore postulate that this bacterium needs to efficiently conserve energy for growth by sensing exterior redox states in its natural habitat. This bacterium may have evolved to use an electrode potential-sensing mechanism comprising the Arc system as an ecological strategy to successfully survive in redox-stratified habitats. Notably, most other members of *Shewanella* lack H$^+$-translocating NDH (Nuo) while they have Na$^+$-translocating NDH (Nqr), suggesting that the regulation of Nuo by the Arc system confers an ecophysiological advantage on MR-1. It is therefore interesting to investigate differences in physiological roles and regulation of these NDHs in *Shewanella*.

Our transcriptome analyses reveal that the Arc system serves as the major signal transduction mechanism that links extracellular redox states to intracellular catabolic pathways. While the expression of genes for many cytoplasm- and IM-localized catabolic enzymes respond to shifts in electrode potential in an ArcS-dependent manner, that of genes encoding EET-pathway proteins (OmcA, MtrCAB, and CymA) was not significantly affected by the potential shift. These EET-pathway proteins are involved in the reduction of extracellular electron acceptors with various redox potentials, including Mn(IV) and Fe(III) oxides[32,46]; therefore, it is reasonable that the genes encoding these proteins are constantly expressed regardless of the redox potential of extracellular substances. Similarly, expression of genes encoding Arc-system proteins (*arcS*, *hptA*, and *arcA*) was not affected by the potential shift. We therefore propose that MR-1 constantly expresses the EET pathway and Arc system for immediate response to changes in extracellular redox states.

A previous study found that the Arc system in MR-1 regulates substantially different genes from those in *E. coli*, even though the amino-acid sequences and target-DNA sequences of ArcA are highly conserved between these bacteria[47]. The same study proposed 54 operons that were likely to be under the direct control of

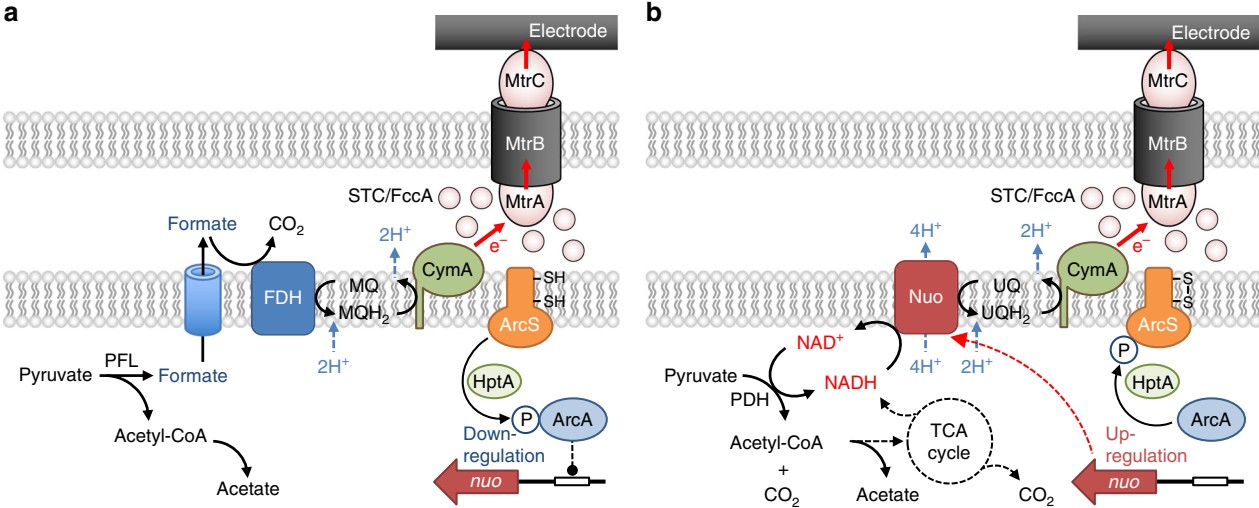

**Fig. 8** Pathways for EET-associated pyruvate catabolism in *S. oneidensis* MR-1. **a** Formate-dependent catabolic pathway consisting of PFL and FDH. **b** NADH-dependent catabolic pathway consisting of PDH and NDH. MQ, oxidized form of menaquinone, MQH$_2$ reduced form of menaquinone, UQ oxidized form of ubiquinone, UQH$_2$ reduced form of ubiquinone

ArcA in MR-1 and concluded that, in contrast to *E. coli*, MR-1 does not use the Arc system for regulating the TCA cycle[48,49]. However, our transcriptome analyses showed that the expression of *sucAB* genes, which encode 2-oxogultarate dehydrogenase, is regulated in an ArcS-dependent and potential-responsive manner. Although MR-1 is known to express an incomplete TCA cycle due to the depressed activity of 2-oxoglutarate dehydrogenase under anaerobic conditions[27], the present finding indicates that MR-1 partially activates the TCA cycle by upregulating expression of *sucAB* genes during respiration using HP electron acceptors. However, our EMSA analysis demonstrates that ArcA does not bind to the upstream region of *sucA* (Supplementary Figure 9). Along with a similar result reported previously[47], this indicates that the Arc system is indirectly involved in the transcriptional regulation of *sucAB* genes. It would therefore be interesting to identify a transcriptional cascade that regulates the expression of these TCA-cycle genes in MR-1.

In contrast to *sucAB* genes, we propose that expression of *nuo* genes is directly regulated by ArcA (Fig. 7). Although *nuo* genes were not included in a previously proposed ArcA regulon in MR-1[47], the expression of these genes is known to be directly regulated by ArcA in *E. coli*[49,50]. Therefore, similar to *E. coli*, MR-1 may use the Arc system to regulate the activity of IM respiratory chains in response to the redox state of IM-associated quinones. However, the expression patterns of *nuo* genes (Supplementary Figures 6 and 7) appear to be inconsistent with the fact that phosphorylated ArcA acts as a repressor for catabolic genes[50]. This can be explained by the fact that ArcA in *E. coli* is autophosphorylated by acetyl phosphate[51], and that its cognate sensor kinase ArcB is involved in the dephosphorylation of ArcA[52]. It is therefore possible that ArcA in MR-1 is also autophosphorylated by similar in vivo non-specific phosphate sources, and that the phosphorylated ArcA represses the expression of its target genes in Δ*arcS*. Further studies are necessary to further our understanding of the roles of the Arc system for catabolic regulation in MR-1.

Among the EET-pathway proteins in MR-1, CymA is required for the reduction of various electron acceptors, including fumarate, nitrate, dimethylsulfoxide, Mn(IV) and Fe(III) oxides[32,53,54], and potential-controlled electrodes in BES (Supplementary Figure 4), indicating that this protein consistently serves as a quinol oxidase under anaerobic conditions, regardless of the redox

potential of electron acceptors. This is in contrast to the fact that *G. sulfurreducens* utilizes different IM-localized cytochromes depending on the redox potential of extracellular electron acceptors[14,15]. Interestingly, quinone analysis (Fig. 4) revealed that the HP electrode oxidizes UQ-8, suggesting that CymA oxidizes ubiquinone when cells are grown under HP conditions. However, oxidation of UQ-8 by CymA is a thermodynamically unfavorable reaction because the potential window of CymA is estimated to range between −0.3 and 0 V (at pH 7.0)[55] compared to the standard redox potential of UQ-8 of +0.11 V[37]. It is therefore hypothesized that this oxidation reaction proceeds only when CymA is almost completely oxidized in the presence of an HP electron acceptor, allowing efficient energy conservation via the NADH-dependent catabolic pathway. A previous study has demonstrated an electrochemical interaction between ubiquinone-10 and CymA[39], supporting the idea that electron transfer from ubiquinone to CymA is possible depending on the in vivo redox balance of these molecules.

In conclusion, the present study demonstrates that the representative EAB, *S. oneidensis* MR-1, has a molecular mechanism for sensing electrode potentials and regulating catabolic pathways. Knowledge of electrode potential-dependent catabolic regulation provides the molecular basis for biotechnological application of EAB. In BES, EAB have been used for current generation and/or chemical synthesis under electrochemical interactions with electrodes[1]; however, in most cases, catabolic pathways in EAB have not been controlled or optimized. Improved understanding of the regulatory mechanisms of catabolic pathways that function in EAB will therefore facilitate the development of more efficient BES. In addition, our findings offer molecular insight into ecological strategies used by EAB in their natural habitats[56,57]. EAB thrive in redox-stratified ecosystems, where they have evolved versatile respiratory abilities[58]. Our findings also suggest that they have evolved redox-dependent regulatory systems that control catabolic pathways. Future studies are required to identify the sensory mechanisms that function in co-operation with the Arc system as cascades for regulating catabolic pathways in response to external redox states.

## Methods

**Bacterial strains and growth conditions**. The bacterial strains and plasmids used in this study are listed in Supplementary Table 1. *E. coli* strains[59] were routinely

cultured in lysogeny broth (LB) medium at 37 °C. *Shewanella oneidensis* strains were cultured at 30 °C in LB medium or minimal medium (MM)[60] containing 10 mM lactate (LMM) or pyruvate (PMM) as the carbon source under aerobic or anaerobic nitrate-, fumarate-, or MnO₂-reducing conditions. For aerobic cultivation, 5 ml medium in a test tube (30 ml capacity) was inoculated with an *S. oneidensis* strain at an initial optical density at 600 nm (OD₆₀₀) of 0.05 and was shaken at 180 rpm. For anaerobic cultivation with nitrate or fumarate, 8 ml PMM supplemented with 2.5 mM nitrate or 20 mM fumarate in a test tube (9 ml capacity) was inoculated with an *S. oneidensis* strain at an initial OD₆₀₀ of 0.01 and was incubated without shaking. Test tubes containing the anaerobic cultures were capped with butyl rubber septa and polycarbonate screw caps, and purged with pure nitrogen gas. The OD₆₀₀ of cultures was measured using a DU800 spectrophotometer (Beckman Coulter, Brea, CA, USA) or a mini photo 518R photometer (Taitec, Tokyo, Japan). For anaerobic cultivation with MnO₂, 80 ml PMM supplemented with 20 mM MnO₂ in a bottle (100 ml capacity) was inoculated with an *S. oneidensis* strain at an initial OD₆₀₀ of 0.01. The bottles containing the cultures were capped with Teflon-coated butyl rubber septa, sealed with aluminum crimp seals, and purged with pure nitrogen gas. MnO₂ powders were prepared according to a method described elsewhere[61]. The amount of MnO₂ in cultures was quantified using a colorimetric assay with leucoberbelin blue[62]. When necessary, 15 µg/ml gentamicin (Gm) or 50 µg/ml kanamycin (Km) was added to culture media.

**Operation of BES.** A small single-chambered three-electrode EC (18 ml total capacity) was used to monitor electric current generated by WT, ΔPDH, ΔPFL, and ΔNDH strains of *S. oneidensis* under potential-controlled conditions. The EC was equipped with a graphite felt working electrode (2.25 cm²), Ag/AgCl reference electrode (+0.199 V vs. SHE) (HX-R5, Hokuto Denko, Tokyo, Japan), and platinum wire counter electrode (10 cm, φ0.3 mm; Nilaco, Tokyo, Japan). The EC was filled with 15 ml LMM supplemented with 170 mM NaCl as an electrolyte, and inoculated with bacterial cells at an initial OD₆₀₀ of 0.1. Current was monitored using a multichannel potentiostat (VPM3; Biologic, Claix, France), and current density (µA/cm²) was calculated based on the projected area of the working electrode. To determine the bacterial growth yield in EC samples, total protein content of planktonic and working electrode-attached cells was determined after the complete consumption of lactate, according to a method described elsewhere[60] with slight modifications. Briefly, the electrolyte (planktonic cell suspension) and working electrode were collected into a 50-ml conical tube and vortexed for 1 min to transfer the bacterial cells attached to the working electrode into the electrolyte. The cell suspension (1 ml) containing the planktonic and electrode-attached cells was mixed with 3 ml of B-PER II bacterial protein extraction reagent (Thermo Fisher Scientific, Walthman, MA, USA) to solubilize the proteins according to the manufacturer's instructions. After centrifugation at 15,000 × *g* for 5 min, the supernatant was collected and used for the protein assay. Protein concentration was determined using a Micro bicinchoninic acid protein assay kit (Thermo Fisher Scientific), and the total protein content in EC was calculated based on a supernatant volume of 15 ml. The protein yield based on lactate (% w/w) in EC was calculated by dividing the total protein content by the amount of lactate consumed.

A large single-chambered EC (180 ml total capacity) was used to analyze current generation coupled to transcriptomics of WT and Δ*arcS* strains. This EC was equipped with a graphite felt working electrode (10 cm²), Ag/AgCl reference electrode, and platinum wire counter electrode (15 cm). The EC was filled with 150 ml LMM supplemented with 170 mM NaCl, and inoculated with bacterial cells at an initial OD₆₀₀ of 0.1. The working electrode was poised at +0.2 V until the electric current became stable (for approximately 15 h) before shifting the potential to +0.5 V or −0.1 V. After further cultivation for 2 h, bacterial cells attached to the working electrode were collected and immediately subjected to RNA extraction. Typical current vs. time curves for WT and Δ*arcS* are shown in Supplementary Figure 1.

**Metabolite analysis.** After the cells were removed by filtration through a membrane filter unit (0.20 µm pore size, DISMIC-25HP; Advantec, Tokyo, Japan), the amount of lactate, acetate, formate, and some other organic acids in the EC supernatant was measured using high-performance liquid chromatography (HPLC; Agilent 1100 series) as described elsewhere[59]. An acetate yield based on the amount of lactate consumed (% mol/mol) in EC was calculated by dividing the amount of acetate produced by the amount of lactate consumed.

**Gene disruption.** In-frame gene-deletion mutants of *S. oneidensis* MR-1 were generated using a two-step homologous recombination method with suicide plasmid pSMV-10 as described previously[60,63]. Briefly, a 1.6-kb fusion product, consisting of upstream and downstream sequences of a target gene joined by an 18-bp linker sequence, was constructed by PCR. Primers for amplifying the flanking regions of the target genes, SO_0577 (*arcS*), SO_2912 (*pflB*), SO_0424 (*aceE*), SO_1017 (*nuoF*), SO_3517 (*ndh*), SO_0907 (*nqrF-1*), and SO_1108 (*nqrF-2*), are listed in Supplementary Table 2. The amplified fusion product was ligated into the *Spe*I site of pSMV-10, and the resultant plasmid was introduced into MR-1 by filter mating with *E. coli* WM6026. Transconjugants (single-crossover clones) were selected on LB plates containing 50 µg/ml kanamycin (Km) and were further

cultivated for 20 h in LB medium lacking antibiotics. The cultures were then spread onto LB plates containing 10% (w/v) sucrose to isolate Km-sensitive double-crossover mutants. Disruption of the target gene in the obtained strains was confirmed by PCR.

**RNA extraction.** Total RNA was extracted using Trizol reagent (Invitrogen) and subsequently purified using an RNeasy Mini Kit and an RNase-Free DNase Set (Qiagen, Valencia, CA, USA) according to the manufacturers' instructions. The quality of the extracted RNA was evaluated using an Agilent 2100 Bioanalyzer with RNA 6000 Pico reagents and RNA Pico Chips (Agilent Technologies).

**qRT-PCR.** qRT-PCR was performed using a LightCycler 1.5 instrument (Roche, Indianapolis, IN, USA) according to a previously described method[64]. Briefly, a PCR reaction mixture contained 15 ng total RNA, 1.3 µl of 50 mM Mn(OAc)₂ solution, 7.5 µl of LightCycler RNA Master SYBR Green I (Roche), and 0.15 µM primers listed in Supplementary Table 2. To generate standard curves, DNA fragments of target genes were amplified by PCR using Ex Taq DNA polymerase (Takara, Tokyo, Japan) and the primer sets listed in Supplementary Table 2, and purified by gel electrophoresis using a QIAEX II Gel Extraction Kit (Qiagen) according to the manufacturer's instructions. Standard curves were generated by amplifying a dilution series of the purified DNA fragments of each gene. The specificity of quantitative PCR was verified by a dissociation-curve analysis. Expression levels of target genes were normalized to the expression level of the 16S rRNA gene.

**Transcriptome analysis.** Transcriptome analysis was performed using custom DNA microarrays (8 × 15K; Agilent Technologies) previously designed based on the annotated genome sequences of *S. oneidensis* MR-1[65] according to the manufacturer's protocol for gene expression arrays for prokaryotes (Agilent One-Color Microarray-Based Prokaryote Analysis, version 1.4, http://www.chem.agilent.com). For transcriptome analysis of the WT strain, five biological replicates were prepared from independent ECs (*n* = 5). For the analysis of Δ*arcS*, three biological and two technical replicates were analyzed (*n* = 6). Gene expression data were normalized and statistically analyzed using GeneSpring GX version 11.5 (Agilent Technologies). Paired Student's *t* test and Benjamini−Hochberg false discovery rate correction were used for statistical analysis. Differential expression for each probe was considered statistically significant when the fold change (FC) was ≥2.0 or ≤0.5 (|log₂ FC| ≥ 1.0) at a *P* value of <0.05. The reliability of the microarray analysis was validated by qRT-PCR of six selected genes (Supplementary Figure 2). There was a high correlation (*r*² = 0.92) between the microarray and qRT-PCR results. Hierarchical clustering was performed using the Cluster 3.0 program[66] and visualized using Java Treeview software[67].

**Measurement of the NADH/NAD⁺ ratio.** A small single-chambered EC was inoculated with MR-1 or ΔNDH cells (precultured anaerobically in LB supplemented with 15 mM lactate and 30 mM fumarate) at an initial OD₆₀₀ of 1.0, and was kept for 4 h at a working electrode potential of +0.5 V, +0.2 V, or 0 V. Bacterial cells were collected from the electrolyte and working electrode using the method described above, and their intracellular NADH/NAD⁺ ratios were determined using an NAD⁺/NADH assay kit (Colorimetric; Abcam, San Francisco, CA, USA) according to the manufacturer's instructions.

**Extraction and analysis of quinones.** A small single-chambered EC was inoculated with MR-1 cells at an initial OD₆₀₀ of 1.0 and was kept for 12 h at a working electrode potential of +0.5 V or −0.1 V. Quinones were extracted from planktonic cells and working electrode-attached cells according to a previously described method with modifications[68,69]. Briefly, 3 ml cell suspension containing planktonic and electrode-attached cells was mixed with the same volume of ice-cold quenching buffer (10 mM tricine in 60% methanol). The cell suspension was transferred to a new amber glass vial and mixed with 3 ml of petroleum ether. After agitating for 1 min, the mixture was centrifuged at 900 × *g* for 2 min, and the upper petroleum ether phase was transferred to a new amber glass vial. Another 3 ml petroleum ether was added to the lower phase, and the agitation and centrifugation steps were repeated. The upper phases were combined and evaporated to dryness under a stream of nitrogen. The dried samples were dissolved in 120 µl of a mixture of methanol and 2-propanol (4:1, v/v) and subjected to HPLC analyses.

The extracted quinone/quinol mixture was analyzed using an HPLC system (Prominence series, Shimadzu, Kyoto, Japan) equipped with a L-column2 ODS column (2.1 × 100 mm, 3.0 µm; CERI, Tokyo, Japan). Isocratic elution of quinones and quinols was achieved by chromatography using the mobile phase consisting of methanol−ethanol (75:25, v/v) at an elution rate of 0.2 ml/min. Qualitative and quantitative analysis was performed using a photodiode array detector and ultraviolet (UV) detector (Shimadzu), respectively. The wavelength of the UV detector was set at 290 nm. UQ-8 (coenzyme Q8) used as a standard was purchased from Avanti Polar Lipids (Alabaster, AL, USA).

**Purification of ArcA protein.** To construct a plasmid expressing ArcA protein with a histidine tag at the N-terminal (N-his-ArcA), the *arcA* gene was amplified

from total DNA of MR-1 using primers arcA_NdeI_F and arcA_BamHI_R (Supplementary Table 2). The resultant PCR products were digested with NdeI and BamHI, and cloned between the corresponding sites of the expression vector pET-28a(+) (Merck, Darmstadt, Germany). The resultant plasmid, pET-arcA (Supplementary Table 1), was introduced into E. coli BL21 (DE3). Cells carrying the plasmid were grown in 300-ml baffled Erlenmeyer flasks containing 100 ml 2× yeast extract-trypton (YT) medium supplemented with Km at 30 °C. Isopropyl-1-thio-β-D-galactopyranoside (final concentration 0.5 mM) was added when the $OD_{600}$ reached 0.5 to 0.8. After overnight cultivation at 25 °C, the cells were harvested by centrifugation, and N-his-ArcA produced by the cells was extracted[70] and purified using a QuickPick IMAC Metal Affinity Kit for Proteins (Bio-Nobile, Turku, Finland) following the manufacturer's instructions. The purified protein samples were analyzed by sodium dodecyl sulfate-polyacrylamide gel electrophoresis (SDS-PAGE) and immediately used for subsequent experiments.

**Electrophoretic mobility shift assay**. The EMSA was performed as previously described with following slight modifications[47,70]. Cy3-labeled DNA probes were generated by PCR using the Cy3-labeled primer sets listed in Supplementary Table 2. Purified N-his-ArcA protein was phosphorylated in buffer containing 100 mM Tris/HCl (pH 7.0), 10 mM $MgCl_2$, 125 mM KCl, 50 mM dilithium carbamoyl phosphate (Sigma-Aldrich, St. Louis, MO, USA) for 60 min at 30 °C. DNA-binding reactions were performed in 12 μl of reaction mixture containing 100 mM Tris-HCl (pH 7.4), 10 mM $MgCl_2$, 20 mM KCl, 2 mM dithiothreitol, 50 μg/ml poly (deoxyinosinic-deoxycytidilic) acid (poly(dI-dC); Sigma-Aldrich), 10% (v/v) glycerol, 4 nM Cy3-labeled DNA probe, and 0−300 ng of phosphorylated N-his-ArcA. The mixture was incubated at 16 °C for 1 h and loaded onto a non-denaturing 12.5% polyacrylamide gel. Electrophoresis was conducted at 150 V in 0.5× Tris-borate-ethylenediaminetetraacetate buffer. Fluorescent gel images were obtained using a Typhoon FLA 9000 (GE Healthcare, Milwaukee, WI, USA).

**Statistical analysis**. Data were statistically evaluated using one-way analysis of variance (ANOVA) followed by Fisher's least significant difference (LSD) test using js-STAR 2012 (http://www.kisnet.or.jp/nappa/software/star/). Differences were considered statistically significant at a P value of <0.05.

**Data availability**. The authors declare that all data supporting the findings of this study are available within the article and its Supplementary Information Files or from the corresponding author upon reasonable request. The microarray data obtained in this study have been deposited in the NCBI Gene Expression Omnibus (GEO) under the accession number GSE102264.

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

## Acknowledgements

This work was supported by JSPS KAKENHI Grant Numbers 17J05454, 24880030, and 26850056. We thank Nanako Amano for technical assistance. We thank Akihiro Okamoto and Kenneth H. Nealson for providing us with the ΔcymA mutant strain of S. oneidensis. We also thank Kenneth H. Nealson for helpful comments and discussions.

## Author contributions

A.K. and K.W. conceived and designed the study; A.H. performed the majority of the experimental work; T.K. contributed to the EMSA experiments; M.A. and T.U. contributed to the quinone analysis; A.K., A.H. and K.W. wrote the paper. All authors discussed the results and commented on the manuscript.

## Additional information

**Competing interests:** The authors declare no competing interests.

