## [Peer Review File · Nature Communications]

Reviewers' comments:

Reviewer #1 (Remarks to the Author):

A number of recent studies has shown that the metabolism of electrochemically active bacteria (EAB) is altered in response to differences in in electrode potentials of electrodes in bioelectrochemical systems. In this study, the authors explored the response of the EAB *Shewanella oneidensis* MR-1 to differences in electrode potentials. Using a number of physiological and transcriptional approaches on various mutants, they show that MR-1 shifts towards using NADH-dependent catabolic pathways which allows to obtain a higher growth yield. The authors also show that MR-1 that the Arc two-component system determines the redox state of the cell and directly or indirectly regulates expression of appropriate genes encoding the corresponding respiratory systems.

The manuscript is very clear, the design of the study is straight forward, and the authors conducted an impressive set of experiments to identify different aspects of the cellular responses. In my view, the results obtained justify the conclusions drawn. Generally, metabolic adaptation to differences in the surrounding electron potential has been demonstrated in numerous examples. In this study, the authors uncover the response of one bacterial species, *S. oneidensis* MR-1 in more detail and demonstrate the involvement of a regulatory system, which is long known for its role in determining the cell redox state.

Apart from that, my general issues with this manuscript are rather minor.

A general remark; when I checked I found that, remarkably, only *S. oneidensis* MR-1 harbors the Nuo H⁺-translocating NADH oxidoreductase. It seems absent in all other *Shewanella* species which rather rely on two Na⁺-translocating Nqr complexes, both of which are also present in MR-1. Thus, part of the findings might be limited to just this single species. On the other hand, I think this is quite remarkable and this aspect should be discussed.

95, how was the significance tested?

138, is that caused by more pyruvate converted into formate/CO₂ under these conditions (less acetate)?

152-154, the results suggest that NADH-dependent pathways are affecting the metabolism at low potential, which was demonstrated before. How does the results obtained for MR-1 relate and

compare to the studies mentioned above (see line 122)? Please clarify or discuss in more detail.

167, how is general growth/protein production of this mutant?

222, how are 'EPR genes' defined?

264, so why is *S. oneidensis* not always using that pathway? Is that due to differences in red/ox states of the quinone pool? Please discuss in more detail.

Minor points

16, please rephrase 'biotechnological attention'.

25/26, do the authors really mean that the Arc regulatory system encompasses extracellular electron pathways and membrane-associated quinones?

35, I suggest to omit 'recently', this has been around for a while.

68, they = the bacteria?

156, (Δ omcAmtrA), insert slash.

171/Fig. S4, is the NADH/NAD⁺ ratio really 0 in LP wt cells?

360, LB is lysogeny broth.

Reviewer #2 (Remarks to the Author):

The paper by Hirose et al. Is a genomic study describing transcriptional response of *Shewanella oneidensis* MR-1 to growth in bioelectrochemical cell under low and high potential electrode settings.

The Watanabe group has previously made significant contribution to understanding extracellular microbial electron transfer and contributed to our current understanding of exo-electrogen physiology and behavior in bioelectrochemical systems.

The study submitted to Nature Communications aims to address important questions that pertain to the mechanisms of redox sensing by *Shewanella* sp., which is central to our predictive understanding and ability to control BES systems. Specifically, the authors hypothesized that “electrochemically active bacteria (or EAB) harbor molecular mechanisms for sensing electrode potentials and regulating catabolic pathways.” Through the application of DNA microarray technology, the transcriptomes of *S. oneidensis* MR-1 wild-type and ArcS deletion mutant were compared under “low potential” (0 mV) and “high-potential” (+500 mV) conditions. Based on observed changes in gene expression patterns, that included a variety of genes involved in electron transport and carbon catabolism, the authors concluded that “MR-1 has molecular mechanisms for electrode potential-dependent catabolic regulation. The findings presented here not only provide the molecular basis for the biotechnological application of EAB in BES but also offer molecular insights into the ecological strategies of EAB in their natural habitats.”

While excited about system and research questions posed by the authors, my enthusiasm was curbed significantly after a close review of the manuscript. There were several major concerns that emerged.

1) **EXPERIMENTAL METHODS:** while the technical direction is straightforward and takes advantage of a large arsenal of genomic information and genetic tools developed for *Shewanella*, it is not clear why the study used only two conditions. It seems that a higher resolution with regard to electrode potential can be substantially more informative and provide a dynamic picture of the organism acclimation. Why weren't the potential chosen to mimic those of the known electron acceptor pairs? There is a tremendous amount of data (including global transcriptomic) accumulated that would have made the comparisons straightforward.

Furthermore, in this day and age, what is the justification for using microarray technology and not taking advantage of next-generation sequencing platform which provides much higher accuracy and ability for de novo discovery.

Lastly, there is nothing on the experimental replication and the potential variability of the microarray data. Statistical analysis is very important here to demonstrate method reproducibility. What about the sample heterogeneity? Were the cells growing as biofilm attached to the electrode?

2) **RESULTS/DISCUSSION:** while the microarray transcript measurements provided a significant amount of data to delve into, the authors chose to focus on comparing their results with previous studies, instead of delivering on their promise to identify “molecular mechanisms for sensing electrode potentials”. To me this is actually the major shortcoming of the manuscript that made its outcomes largely confirmatory in nature without significantly advancing our

knowledge of redox sensing.

For example: the roles of PDH and PFL are well-established in *S. oneidensis* MR-1 and inactivation of the former will result in growth/energy production deficiency under anaerobic (low potential conditions). On the other hand PDH is an enzyme that is active under aerobic conditions and will be dominant under microaerobic (high potential) conditions.

Also, it appears that there is some confusion on the authors' part with regard to the mechanisms of energy conservation (ll. 118-122) and the comparison drawn at high potential conditions. First, numerous studies established that MR-1 generates reductant and ATP through oxidative phosphorylation under both aerobic and anaerobic conditions. There are several substrate-level phosphorylation steps for ATP regeneration as well, which can be carried out simultaneously with oxidative phosphorylation. Second, it is not correct to refer to high potential (+500 mV) conditions as strict anaerobic. This is more along the lines of microaerobic/oxygen-limited growth; clearly, additional redox conditions would have benefited the analysis (see comment 1).

Finally, what is the logic behind focusing on the arc regulatory system? While it has been shown to modulate the expression of the aerobic metabolism of MR-1, there are other global regulators that affect anaerobic electron transfer to a significantly larger extent. For instance, inactivation of the cAMP receptor protein (CRP) was shown to abolish the expression of many terminal reductases. Why mutants deficient in these global regulators were not analyzed?

Reviewer #3 (Remarks to the Author):

The authors present a comprehensive study on the use of a respiratory pathway to an anode by *Shewanella oneidensis* MR-1. As the authors discuss, the generation of electrical current by this microorganism has been debated for over a decade since it is not clear whether *Shewanella* can gain energy from this process. Given that the lactate fermentation produces ATP, *Shewanella* might produce current as a way to get rid of excess electron equivalents. Through a combination of electrochemical, transcriptomics, and microbiological techniques, the authors confirm that *S. Oneidensis* MR-1 does utilize the anode through a respiratory pathway that generates ATP. I think the manuscript provides an important contribution to the several research fields and it is of high importance in the general understanding of this microorganism.

My main concerns are related to the discussion of the data or its interpretation, as outlined below. I hope my comments will help the authors improve the delivery of their message to the scientific

community:

1. The main goal of the study is to determine whether MR-1 will generate ATP from its current generation, as shown in Fig. 8. An alternative is that the delivery of electrons is not associated with energy conservation, but as a way to get rid of excess electrons. In either case, a more positive anode potential will result in a faster and more efficient way to produce electrical current. Therefore, an increase in current density, a more oxidized quinone pool, and even a faster growth can be expected of either possibility. Because of this, Figure 1c is the crucial data set since it shows a higher yield, associated with the ability to assimilate more substrate. In my opinion, the authors should emphasize the importance of Figure 1c and discuss why this is only possible if energy conservation is increased with increasing anode potential.

2. I disagree with the statement in ln 42 because it is too general:

“However, it is unclear whether or not EAB can actively utilize electrode potentials for energy conservation.”

Microorganisms like *G. sulfurreducens* grow exclusively on respiration to an anode, unlike *Shewanella* that can obtain ATP from lactate fermentation to acetate (Fig 2). Therefore, it is too general to say that it is unclear whether this happens in all EAB. In line 70, the authors correctly refers to MR-1 on a similar concept:

“Despite extensive characterization of electron-transfer pathways, however, it remains unclear how redox potentials of electron acceptors affect the growth of MR-1.”

I would make sure to refer to MR-1 and not all EAB in the first statement.

3. I also disagree with the following statement on ln 89: “The HP condition produced higher electric currents than the LP condition (Fig. 1a), indicating that the HP electrode facilitates cellular catabolic reactions, such as lactate oxidation and EET.” First, EET should not be considered a catabolic reaction. Second, the higher current density on its own only shows a higher activity of electron flow to the anode and does not indicate any direct correlation to the cell’s catabolism.

I do agree that the increase in current is an important measurement and part of the evidence that relates current generation to ATP generation and faster growth. However, this evidence on its own cannot be used to conclude it.

4. The discussion on the Arc system seems to take over the latter part of the discussion and conclusion. While interesting, I am not convinced the data entirely supports the hypothesis of the Arc system as a regulatory, potential sensing system. The evidence is not as compelling as the rest of the story just because it is not as comprehensive. It is nonetheless an important point of discussion, but I will suggest that the manuscript deemphasizes this story or puts a bit less focus on it.

If the authors decide to keep a wider focus on the Arc System, I would suggest including it on the schematic shown on Figure 8.

Responses to Reviewers' Comments:

Responses to Reviewer #1:

We thank the Reviewer for carefully reading our manuscript and making many valuable comments and suggestions that have helped us significantly improve our paper. As indicated below, we have taken all these comments and suggestions into account in the revised version of our manuscript.

Comment 1: *A number of recent studies has shown that the metabolism of electrochemically active bacteria (EAB) is altered in response to differences in electrode potentials of electrodes in bioelectrochemical systems. In this study, the authors explored the response of the EAB Shewanella oneidensis MR-1 to differences in electrode potentials. Using a number of physiological and transcriptional approaches on various mutants, they show that MR-1 shifts towards using NADH-dependent catabolic pathways which allows to obtain a higher growth yield. The authors also show that MR-1 that the Arc two-component system determines the redox state of the cell and directly or indirectly regulates expression of appropriate genes encoding the corresponding respiratory systems.*

The manuscript is very clear, the design of the study is straight forward, and the authors conducted an impressive set of experiments to identify different aspects of the cellular responses. In my view, the results obtained justify the conclusions drawn. Generally, metabolic adaptation to differences in the surrounding electron potential has been demonstrated in numerous examples. In this study, the authors uncover the response of one bacterial species, S. oneidensis MR-1 in more detail and demonstrate the involvement of a regulatory system, which is long known for its role in determining the cell redox state.

Apart from that, my general issues with this manuscript are rather minor.

Response: We thank the Reviewer for highly evaluating our manuscript. We hope that the following responses are satisfactory.

Comment 2: *A general remark; when I checked I found that, remarkably, only S. oneidensis MR-1 harbors the Nuo H⁺-translocating NADH oxidoreductase. It seems absent in all other Shewanella species which rather rely on two Na⁺-translocating Nqr complexes, both of which are also present in MR-1. Thus, part of the findings might be limited to just this single species. On the other hand, I think this is quite remarkable and this aspect should be discussed.*

Response: We thank the Reviewer for this valuable comment. We also checked the presence of Nuo in *Shewanella* using the KEGG and NCBI nr databases. We found that several *Shewanella* species, including *S. violacea*, *S. woodyi*, and *S. psychrophila*, have putative NDHs that show relatively high homology with Nuo in MR-1 (>55% amino acid identity), although their exact functions are not clear. As the Reviewer indicated, however, most other members of *Shewanella* do not have Nuo. We have added sentences about this discussion (line 259–299).

Comment 3: 95, how was the significance tested?

Response: We apologize for the insufficient description. The data were statistically evaluated using ANOVA. We have added the “Statistical analysis” subsection in the “Methods” and described the detailed methods for statistical analysis used in this study.

Comment 4: 138, is that caused by more pyruvate converted into formate/CO₂ under these conditions (less acetate)?

Response: Although the present transcriptome analysis revealed that the expression of the three FDH operons is differentially regulated in response to the electrode potential, the detailed mechanisms underlying their differential expression are currently unclear. We have revised this sentence to more clearly describe the situation (line 140 to 142).

Comment 5: 152-154, the results suggest that NADH-dependent pathways are affecting the metabolism at low potential, which was demonstrated before. How does the results obtained for MR-1 relate and compare to the studies mentioned above (see line 122)? Please clarify or discuss in more detail.

Response: We thank the Reviewer for this insightful comment. As the Reviewer indicated, the Δ PDH mutant generated higher current than WT in the presence of the low potential electrode. Although the reason for this phenomenon is not clear, it is likely that the presence of NADH-dependent pathways negatively affects catabolism under low potential conditions where the formate-dependent catabolic pathway is predominantly utilized. We have added the sentences about this discussion (line 158–160).

Comment 6: 167, how is general growth/protein production of this mutant?

Response: This mutant (Δ NDH) grew under anaerobic fumarate-reducing conditions, but did not grow under aerobic conditions (please see Fig. 4a). This is described in a paragraph starting at line 188.

Comment 7: 222, how are 'EPR genes' defined?

Response: 'EPR genes' indicate 'electrode potential-responsive genes' that were identified from the transcriptome analyses. We defined it as described in line 111.

Comment 8: 264, so why is *S. oneidensis* not always using that pathway? Is that due to differences in red/ox states of the quinone pool? Please discuss in more detail.

Response: We consider that efficient energy conservation via the NADH-dependent catabolic pathway requires electron acceptors with relatively high redox potentials, such as oxygen, high potential electrodes, and MnO_2 , because this process involves the redox cycling of ubiquinone (Fig. 8b) and can proceed only in the presence of these high potential electron acceptors. To state this point more clearly, we have revised the text in the Discussion (line 353–356).

Comment 9: 16, please rephrase 'biotechnological attention'.

Response: We have revised the Abstract.

Comment 10: 25/26, do the authors really mean that the Arc regulatory system encompasses extracellular electron pathways and membrane-associated quinones?

Response: Our results indicate that MR-1 uses the Arc regulatory system to sense the redox state of membrane quinones that link to electrodes via the extracellular electron transfer pathway, thereby capable of responding to changes in electrode potentials. Due to the word limit of the abstract, however, this part was deleted.

Comment 11: 35, I suggest to omit 'recently', this has been around for a while.

Response: In accordance with the Reviewer's suggestion, we have revised the text.

Comment 12: 68, *they = the bacteria?*

Response: ‘they’ means ‘BES (bioelectrochemical systems)’. To avoid a misunderstanding, we have changed ‘they’ into ‘these systems’.

Comment 13: 156, (*ΔomcAmtrA*), *insert slash*.

Response: We have corrected the typographical error.

Comment 14: 171/Fig. S4, *is the NADH/NAD⁺ ratio really 0 in LP wt cells?*

Response: The NADH/NAD⁺ ratio in the WT cells grown under the LP condition was not zero but a very small value. This result is reasonable because the NADH/NAD⁺ ratios gradually increased as the electrode potential increased (please see Supplementary Fig. S4 in the revised manuscript). We consider that this shift is associated with the activation of the NADH-dependent catabolic pathway under high potential conditions.

Comment 15: 360, *LB is lysogeny broth*.

Response: We have corrected the error.

Response to Reviewer #2:

We thank the Reviewer for valuable comments and suggestions that have helped us improve our paper. We hope that the following responses are satisfactory.

Comment 1-1: *EXPERIMENTAL METHODS: while the technical direction is straightforward and takes advantage of a large arsenal of genomic information and genetic tools developed for Shewanella, it is not clear why the study used only two conditions. It seems that a higher resolution with regard to electrode potential can be substantially more informative and provide a dynamic picture of the organism acclimation. Why weren't the potential chosen to mimic those of the known electron acceptor pairs? There is a tremendous*

amount of data (including global transcriptomic) accumulated that would have made the comparisons straightforward.

Response: In the original manuscript, we chose the HP (+0.5 V vs. SHE) and LP (0 V) conditions, because these potentials are close to the redox potentials of MnO₂ and fumarate (line 191–192), respectively, which are both preferentially utilized by *Shewanella* as anaerobic electron acceptors. In addition, this range of potentials (0–0.5 V) are generally used for electrogenic BES and anaerobic cultivation of *Shewanella*. This study therefore focuses on the regulation of catabolic pathways within this potential range to investigate how *S. oneidensis* MR-1 regulates catabolic pathways and conserves energy during anaerobic respiration using electrodes and natural electron acceptors. However, we understand the Reviewer's concern that the addition of more potential conditions can provide more information, and have added the middle potential (MP; +0.2 V) condition (Fig. 1, 3, and Supplementary Fig. S4) and modified the relevant texts in the revised manuscript. The additional data indicate that MR-1 shows moderate responses to the MP conditions as compared to the HP and LP conditions, supporting that MR-1 activates the NADH-dependent catabolic pathway when it utilizes electron acceptors with high redox potentials. Together with the results obtained in the experiments using different electron acceptors with different potentials (ranging between +0.8 V and 0 V vs. SHE), our work could provide with sufficient data for discussing relationships between redox potentials and catabolic pathways.

Comment 1-2: *Furthermore, in this day and age, what is the justification for using microarray technology and not taking advantage of next-generation sequencing platform which provides much higher accuracy and ability for de novo discovery.*

Response: In this study, we focused on detecting differential expression of the annotated genes in the well-characterized *S. oneidensis* MR-1. For such purpose, microarray technology can yield reliable and reproducible results with sufficient sensitivity and accuracy. Many studies recently published in *Nature communications* also employ the microarray technology (e.g., Georgiev et al. 2016. doi: 10.1038/ncomms13116; Zaytouni et al. 2017. doi: 10.1038/s41467-017-00331-y). We would also like the Reviewer to understand that, at least in Japan, RNA sequencing is more expensive and time-consuming than DNA microarray.

Comment 1-2: *Lastly, there is nothing on the experimental replication and the potential variability of the microarray data. Statistical analysis is very important here to demonstrate method reproducibility. What about the sample heterogeneity? Were the cells growing as biofilm attached to the electrode?*

Response: We prepared at least three biological replicates (i.e., independent electrochemical reactors) for the present microarray experiments, and the data were statistically analyzed by paired Student's t-test and Benjamini-Hochberg false discovery rate correction. Differential expression for each probe was considered statistically significant when the fold change was ≥ 2.0 or ≤ 0.5 at a *P*-value of < 0.05 . The reliability of the microarray analysis was validated by qRT-PCR of six selected genes (Supplementary Fig. S2), and there was a high correlation ($r^2 = 0.92$) between the microarray and qRT-PCR results. The microarray data have been deposited in the NCBI Gene Expression Omnibus (GEO) according to the MIAME guideline. We therefore consider that the microarray data in the present study are highly reproducible and reliable and meet the standard criteria. As shown in Fig. 1, bacterial cells can grow after the electrochemical cultivation. In the microarray experiments, however, we collected bacterial cells two hours after the potential shift (Supplementary Fig. S1) to minimize a difference in cell growth between the HP and LP conditions.

Comment 2-1: *RESULTS/DISCUSSION: while the microarray transcript measurements provided a significant amount of data to delve into, the authors chose to focus on comparing their results with previous studies, instead of delivering on their promise to identify “molecular mechanisms for sensing electrode potentials”. To me this is actually the major shortcoming of the manuscript that made its outcomes largely confirmatory in nature without significantly advancing our knowledge of redox sensing.*

*For example: the roles of PDH and PFL are well-established in *S. oneidensis* MR-1 and inactivation of the former will result in growth/energy production deficiency under anaerobic (low potential conditions). On the other hand PDH is an enzyme that is active under aerobic conditions and will be dominant under microaerobic (high potential) conditions.*

Response: In the present study, we focus on how *S. oneidensis* MR-1 regulates catabolic pathways and conserve energy in response to changes in the electrode potential. As Reviewer #3 indicated, this question remains to be addressed even in this well-characterized, model electricity-generating bacterium. In lactate catabolism by

MR-1, the regulation of PDH/NDH and PFL is critically important because it determines whether the NADH- or formate-dependent pathway is utilized for lactate/pyruvate catabolism (Fig. 2). We therefore describe the manuscript with a particular focus on the regulation of these enzymes (genes). As the Reviewer #2 pointed out, a previous study by Pinchuk et al. (Appl. Environ. Microbiol. 2011, 77:8234) have characterized the physiological roles of PDH and PFL, demonstrating that PDH and PFL are activated under aerobic and O₂-limited conditions, respectively. However, signal transduction mechanisms underlying these regulations remain to be identified. In the present study, we show that MR-1 activates the NADH-dependent catabolic pathway consisting of PDH and NDH under high potential conditions, indicating that the regulation of this pathway is associated with the redox potentials of electron acceptors. This is the first report demonstrating that the electrode potential affects catabolic pathways in *Shewanella*, and we consider that the findings presented herein contain sufficient novelty.

Comment 2-2: *Also, it appears that there is some confusion on the authors' part with regard to the mechanisms of energy conservation (ll. 118-122) and the comparison drawn at high potential conditions. First, numerous studies established that MR-1 generates reductant and ATP through oxidative phosphorylation under both aerobic and anaerobic conditions. There are several substrate-level phosphorylation steps for ATP regeneration as well, which can be carried out simultaneously with oxidative phosphorylation.*

Response: Kristopher et al. (J. Bacteriol. 2010. 192:3345) have reported that substrate-level phosphorylation via acetate synthesis is the primary source of energy conservation during anaerobic respiration (fumarate reduction) by MR-1. We have also obtained similar results using Δ *pta* and Δ *atp* mutants (unpublished data). Reviewer #3 has also commented that 'it is not clear whether *Shewanella* can gain energy from this process (extracellular electron transfer/electrode respiration)'. Although Kane et al. (J. Bacteriol. 2016. 198:1337) have reported that formate oxidation contributes to the generation of proton motive force (PMF) under fumarate-reducing conditions, they have also shown that F-type ATPase is not essential for the growth of MR-1 under anaerobic conditions, and conclude that this enzyme works in the opposite direction to hydrolyze ATP and to generate PMF under anaerobic conditions. It is therefore not conclusive that MR-1 can generate ATP through oxidative phosphorylation under anaerobic conditions.

Comment 2-3: *Second, it is not correct to refer to high potential (+500 mV) conditions as strict anaerobic. This is more along the lines of microaerobic/oxygen-limited growth; clearly, additional redox conditions would have benefited the analysis (see comment 1).*

Response: The potential of +0.5 V is close to that of MnO₂, and thus is considered usual for *Shewanella*. Our results suggest that *S. oneidensis* has evolved to efficiently conserve energy during respiration using anaerobic electron acceptors with relatively high redox potentials, such as MnO₂. This notion is likely because MR-1 was isolated from the sediments of a shallow eutrophic lake containing iron/manganese-oxide concretion at the bottom (Myers and Nealson. 1988. Science 240:1319).

Comment 2-4: *Finally, what is the logic behind focusing on the arc regulatory system? While it has been shown to modulate the expression of the aerobic metabolism of MR-1, there are other global regulators that affect anaerobic electron transfer to a significantly larger extent. For instance, inactivation of the cAMP receptor protein (CRP) was shown to abolish the expression of many terminal reductases. Why mutants deficient in these global regulators were not analyzed?*

Response: Our laboratory has published manuscripts that report roles of CRP in the transcriptional regulation of terminal reductases and catabolic enzymes (Kasai et al. 2015. BMC Microbiol.; Kasai et al. 2017. Front. Microbiol). In the present study, however, the transcriptome analyses demonstrated that most of anaerobic terminal reductase genes whose expression is regulated by CRP were not significantly responsive to changes in the electrode potential (Supplementary Table S3). We therefore considered that another regulator(s) should be involved in electrode potential sensing by MR-1. We hypothesized that the Arc system would play the central role, because the electrode potential affected the redox balance of membrane quinones (Fig. 5), which can be sensed by this regulatory system, and primarily addressed this hypothesis.

Response to Reviewer #3:

We thank the Reviewer for giving many valuable comments and suggestions that have helped us significantly improve our paper. As indicated in the responses that follow, we have taken all these comments and suggestions into account in the revised version of our manuscript.

Comment: *The authors present a comprehensive study on the use of a respiratory pathway to an anode by Shewanella oneidensis MR-1. As the authors discuss, the generation of electrical current by this microorganism has been debated for over a decade since it is not clear whether Shewanella can gain energy from this process. Given that the lactate fermentation produces ATP, Shewanella might produce current as a way to get rid of excess electron equivalents. Through a combination of electrochemical, transcriptomics, and microbiological techniques, the authors confirm that S. oneidensis MR-1 does utilize the anode through a respiratory pathway that generates ATP. I think the manuscript provides an important contribution to the several research fields and it is of high importance in the general understanding of this microorganism.*

My main concerns are related to the discussion of the data or its interpretation, as outlined below. I hope my comments will help the authors improve the delivery of their message to the scientific community:

Response: We thank the Reviewer for evaluating our manuscript. As the Reviewer indicated, we also consider that our results provide evidence that *S. oneidensis* MR-1 can conserve energy by oxidative phosphorylation during anaerobic respiration, particularly when it utilizes electron acceptors with relatively high redox potentials. We hope that the following responses and revisions are satisfactory.

Comment 1: *The main goal of the study is to determine whether MR-1 will generate ATP from its current generation, as shown in Fig. 8. An alternative is that the delivery of electrons is not associated with energy conservation, but as a way to get rid of excess electrons. In either case, a more positive anode potential will result in a faster and more efficient way to produce electrical current. Therefore, an increase in current density, a more oxidized quinone pool, and even a faster growth can be expected of either possibility. Because of this, Figure 1c is the crucial data set since it shows a higher yield, associated with the ability to assimilate more substrate. In my opinion, the authors should emphasize the importance of Figure 1c and discuss why this is only possible if energy conservation is increased with increasing anode potential.*

Response: We thank the Reviewer for this valuable comment. We also understand the importance of Fig. 1c, and proposed a mechanism underlying efficient energy conservation under HP conditions in the first paragraph in the Discussion (line 262–285).

To emphasize this important discussion more clearly, we have modified some sentences in this paragraph.

Comment 2: *I disagree with the statement in ln 42 because it is too general: “However, it is unclear whether or not EAB can actively utilize electrode potentials for energy conservation.” Microorganisms like G. sulfurreducens grow exclusively on respiration to an anode, unlike Shewanella that can obtain ATP from lactate fermentation to acetate (Fig 2). Therefore, it is too general to say that it is unclear whether this happens in all EAB. In line 70, the authors correctly refers to MR-1 on a similar concept: “Despite extensive characterization of electron-transfer pathways, however, it remains unclear how redox potentials of electron acceptors affect the growth of MR-1.” I would make sure to refer to MR-1 and not all EAB in the first statement.*

Response: We used the word “actively” for describing the abilities of bacteria to sense and utilize electrode potentials, and the ability for sensing electrode potentials has not been identified even for *Geobacter*. However, in order to avoid confusion, we revised the relevant sentences in the Abstract and Introduction for more specific description (lines 18 and 39–40).

Comment 3: *I also disagree with the following statement on ln 89: “The HP condition produced higher electric currents than the LP condition (Fig. 1a), indicating that the HP electrode facilitates cellular catabolic reactions, such as lactate oxidation and EET.” First, EET should not be considered a catabolic reaction. Second, the higher current density on its own only shows a higher activity of electron flow to the anode and does not indicate any direct correlation to the cell’s catabolism. I do agree that the increase in current is an important measurement and part of the evidence that relates current generation to ATP generation and faster growth. However, this evidence on its own cannot be used to conclude it.*

Response: We agree that EET is not a catabolic reaction. We also agree that, in some situations, electric current is not linked to intracellular catabolism; for instance, when excess reducing equivalents are accumulated in cells. However, in other situations, electric current occurs with electrons that are continuously supplied by catabolic reactions, and high catabolic activities therefore generate higher electric currents. Taking these discussions together into account, we revised the sentence (line 87–89).

Comment 4: *The discussion on the Arc system seems to take over the latter part of the discussion and conclusion. While interesting, I am not convinced the data entirely supports the hypothesis of the Arc system as a regulatory, potential sensing system. The evidence is not as compelling as the rest of the story just because it is not as comprehensive. It is nonetheless an important point of discussion, but I will suggest that the manuscript deemphasizes this story or puts a bit less focus on it. If the authors decide to keep a wider focus on the Arc System, I would suggest including it on the schematic shown on Figure 8.*

Response: In accordance with the Reviewer's suggestion, we revised Fig. 8. In addition, we agree that further studies are necessary for confirming the roles of the Arc system in MR-1, and some parts of the manuscript were modified to deemphasize the Arc story (line 23 to line 25; line 341 to line 342).

Thank you again for your comments on our paper. We hope that the revised version of our paper is now suitable for publication in *Nature communications*.

Reviewers' Comments:

Reviewer #1 (Remarks to the Author):

The authors have addressed my concerns satisfyingly.

Reviewer #3 (Remarks to the Author):

The authors have addressed my comments with great detail. I think this will be a great publication.

Reviewer #4 wrote in Remarks to Editors section. (S)he thinks the revision has improved the manuscript.